# Estimating the COVID-19 epidemic trajectory and hospital capacity requirements in South West England: a mathematical modelling framework

Ross D Booton ,[1] Louis MacGregor,[2,3] Lucy Vass,[1,2] Katharine J Looker ,[2,3] Catherine Hyams,[4] Philip D Bright,[5] Irasha Harding,[6] Rajeka Lazarus,[7] Fergus Hamilton,[8] Daniel Lawson ,[9] Leon Danon,[2,10,11,12] Adrian Pratt,[13] Richard Wood ,[12,13] Ellen Brooks-Pollock ,[1,2,3] Katherine M E Turner [1,2,3,12]

EB-P and KMET are joint senior authors.

For numbered affiliations see end of article.

**Correspondence to**
Dr Katherine M E Turner;
katy.turner@bristol.ac.uk

## ABSTRACT

**Objectives** To develop a regional model of COVID-19 dynamics for use in estimating the number of infections, deaths and required acute and intensive care (IC) beds using the South West England (SW) as an example case.

**Design** Open-source age-structured variant of a susceptible-exposed-infectious-recovered compartmental mathematical model. Latin hypercube sampling and maximum likelihood estimation were used to calibrate to cumulative cases and cumulative deaths.

**Setting** SW at a time considered early in the pandemic, where National Health Service authorities required evidence to guide localised planning and support decision-making.

**Participants** Publicly available data on patients with COVID-19.

**Primary and secondary outcome measures** The expected numbers of infected cases, deaths due to COVID-19 infection, patient occupancy of acute and IC beds and the reproduction ('R') number over time.

**Results** SW model projections indicate that, as of 11 May 2020 (when 'lockdown' measures were eased), 5793 (95% credible interval (CrI) 2003 to 12 051) individuals were still infectious (0.10% of the total SW population, 95% CrI 0.04% to 0.22%), and a total of 189 048 (95% CrI 141 580 to 277 955) had been infected with the virus (either asymptomatically or symptomatically), but recovered, which is 3.4% (95% CrI 2.5% to 5.0%) of the SW population. The total number of patients in acute and IC beds in the SW on 11 May 2020 was predicted to be 701 (95% CrI 169 to 1543) and 110 (95% CrI 8 to 464), respectively. The R value in SW was predicted to be 2.6 (95% CrI 2.0 to 3.2) prior to any interventions, with social distancing reducing this to 2.3 (95% CrI 1.8 to 2.9) and lockdown/school closures further reducing the R value to 0.6 (95% CrI 0.5 to 0.7).

**Conclusions** The developed model has proved a valuable asset for regional healthcare services. The model will be used further in the SW as the pandemic evolves, and—as open-source software—is portable to healthcare systems in other geographies.

### Strengths and limitations of this study

► Open-source modelling tool available for wider use and reuse.

► Customisable to a number of granularities such as at the local, regional and national levels.

► Supports a more holistic understanding of intervention efficacy through estimating unobservable quantities, for example, asymptomatic population.

► While not presented here, future use of the model could evaluate the effect of various interventions on transmission of COVID-19.

► Further developments could consider the impact of bedded capacity in terms of resulting excess deaths.

## INTRODUCTION

Since the initial outbreak in 2019 in Hubei Province, China, COVID-19, the disease caused by SARS-CoV-2, has gone on to cause a pandemic.[1] As of 11 May 2020, the Centre for Systems Science and Engineering at Johns Hopkins University reports over 4 000 000 confirmed cases and 250 000 deaths globally.[2] National responses to the outbreak have varied: from severe restrictions on human mobility alongside widespread testing and contact tracing in China[3] to the comparatively relaxed response in Sweden, where lockdown measures have not been enacted.[4] In the UK, advice to socially distance if displaying symptoms was given on 15 March, while school closures and 'lockdown' measures were implemented from 23 March onwards.[5]

Mathematical modelling has been used to predict the course of the COVID-19 pandemic and to evaluate the effectiveness of proposed and enacted interventions.[6–11] Prem *et al* [6] showed that the premature lifting of control strategies at the national level (within China) could lead to an earlier secondary peak;

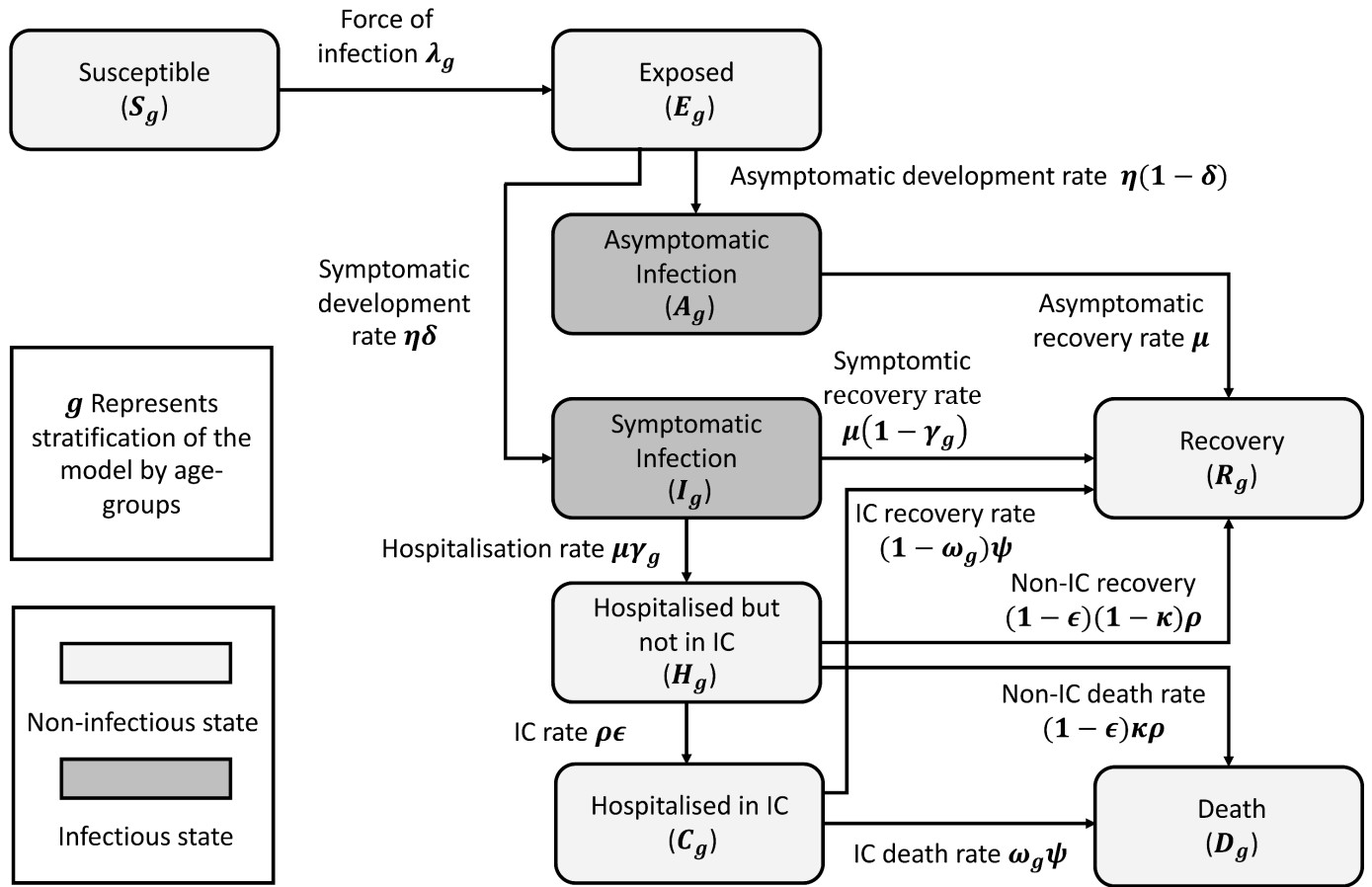

**Figure 1** Compartmental flow model diagram depicting stages of disease and transitions between states. Asymptomatic infection represents the number of people never showing symptoms while symptomatic infection includes all those who show presymptomatic/mild symptoms to those who show more severe symptoms (prehospitalisation). Those who are hospitalised first occupy a non-IC bed (acute bed) after which they can either move into IC, recover or die. Those in IC can either recover or die at an increased rate compared with those in acute beds. This model does not capture those deaths which occur outside of hospital as a result of COVID-19. IC, intensive care.

Flaxman *et al*[7] used a semimechanistic model to predict the total COVID-19 infections in 11 countries; Ferguson *et al*[8] used an individual-based simulation model of COVID-19 transmission to explore the effects of non-pharmaceutical interventions within the USA and Great Britain; Challen *et al*[9] estimated the R number among regions of the UK; Danon *et al*[10] used a spatial model to predict the potential course of COVID-19 in England and Wales in the absence of control measures; while Jarvis *et al*[11] analysed the behavioural monitoring data to quantify the impact of control measures on COVID-19 transmission. These models have been predominantly aimed at the national level and have largely been based on epidemiological and biological data sourced from the initial epidemic in Wuhan, China,[12] and the first large outbreak in Lombardy, Italy.[13] These models have also mainly focused on predicting the scale of COVID-19 transmission under various intervention measures, rather than producing estimates for potential numbers of COVID-19-related admissions to acute or intensive care (IC).

In the UK, the epidemic escalated most rapidly in London,[14] and the majority of national modelling is seemingly driven by the trends in London due to its large case numbers and large population. One of the key issues facing National Health Service (NHS) authorities is planning for more localised capacity needs and estimating the timings of surges in demand at a regional or healthcare system level. This is especially challenging given the rapidly evolving epidemiological and biological data; the changes in COVID-19 testing availability (eg, previously limited and changing eligibility requirements); the uncertainty in the effectiveness of interventions in different contexts; significant and uncertain time lags between initial infection and hospitalisation or death; and different regions being at different points in the epidemic curve.[9] South West England (SW) is the region with the lowest number of total cases in England (as of 11 May 2020), lagging behind the national data driven by the earlier epidemic in London.[9 14]

COVID-19 results in a significant requirement for hospitalisation and high mortality among patients requiring admission to critical care (particularly among those requiring ventilation).[15 16] In the SW, the population is on average older than in London[17] and is older than the UK as a whole (online supplemental table S1). Older age puts individuals at elevated risk of requiring

hospital care.[18–20] Consequently, we might expect higher mortality and greater demand for beds in the SW than estimations output from national models that may lack such granularity or risk sensitivity.

However, the SW's first case occurred around 2 weeks later than the first UK case[14]; perhaps implying that the local SW epidemic may be more effectively controlled due to a lower number of baseline cases (than the national average) at the time national interventions were implemented, as well as reduced transmission due to rurality. This subnational analysis can support in mapping the local epidemic, planning local hospital capacity outside of the main urban centres and ensuring effective mobilisation of additional support and resources if required. Should demand be lower than expected, reliable forecasts could facilitate more effective use of available resources through reintroducing elective treatments (that had initially been postponed) and responding to other non-COVID-19 sources of emergency demand.

In this study, taking into account the timeline of UK-wide non-pharmaceutical interventions (social distancing, school closures/lockdown), we illustrate use of our model in projecting estimates for the expected distributions of cases, deaths, asymptomatic and symptomatic infections and demand for acute and IC beds. We present the model trajectories for SW using publicly available data.

## METHODS

We developed a deterministic, ordinary differential equation model of the transmission dynamics of COVID-19, including age-structured contact patterns, age-specific disease progression and demand for hospitalisation, both to acute and IC. We then parameterised the model using available literature and calibrated the model to data from the SW. The model is readily adapted to fit the data at subregional (eg, Clinical Commissioning Group, CCG), regional or national level. Key assumptions of the model are summarised in the online supplemental information.

The model was developed in R and all code and links to source data are freely available (*github.com/rdbooton/bricovmod*). The model is coded using package *deSolve*, with contact matrices from package *socialmixr* and sampling from package *lhs*.

### Model structure

The stages of COVID-19 included within this model are $S$—susceptible, $E$—exposed (not currently infectious but have been exposed to the virus), $A$—asymptomatic infection (will never develop symptoms), $I$—symptomatic infection (consisting of presymptomatic or mild to moderate symptoms), $H$—severe symptoms requiring hospitalisation but not IC, $C$—very severe symptoms requiring IC, $R$—recovered and $D$—death. The total population is $N = S + E + A + I + H + C + R + D$ (figure 1).

Each compartment $X_g$ is stratified by age group (0–4, 5–17, 18–29, 30–39, 40–49, 50–59, 60–69, ≥70) where $X$ denotes the stage of COVID-19 (S, E, A, I, H, C, R,

D) and $g$ denotes the age group class of individuals. Age groups were chosen to capture key social contact patterns (primary, secondary and tertiary education and employment) and variability in hospitalisation rates and outcomes from COVID-19 especially in older age groups. The total in each age group is informed by recent Office for National Statistics (ONS) estimates.[21]

Susceptible individuals become exposed to the virus at a rate governed by the force of infection $\lambda_g$, and individuals are non-infectious in the exposed category. A proportion $\delta$ move from exposed to symptomatic infection and the remaining to asymptomatic infection, both at the latent rate $\eta$. Individuals leave both the asymptomatic and symptomatic compartments at rate $\mu$. All asymptomatic individuals eventually recover and there are no further stages of disease: the rate of leaving the asymptomatic compartment is therefore equivalent to the infectious period, $\mu$. A proportion of symptomatic individuals $\gamma_g$ go on to develop severe symptoms which require hospitalisation, but not IC. Once requiring hospitalisation, we assume individuals are no longer infectious to the general population due to self-isolation guidelines restricting further mixing with anyone aside from household members (if unable to be admitted to hospital) or front-line NHS staff (if admitted to hospital). Individuals move out of the acute hospitalised compartment at rate $\rho$, either through death, being moved to IC at rate $\epsilon$, or through recovery (all remaining individuals). A proportion $\omega_g$ of patients requiring IC will die at rate $\psi$, while the rest will recover.

The model (schematic in figure 1) is therefore described by the following differential equations:

| | | |
|---|---|---|
| Susceptible $S_g$ | $\dfrac{dS_g}{dt} = -\lambda_g S_g$ | (1a) |
| Exposed $E_g$ | $\dfrac{dE_g}{dt} = \lambda_g S_g - \eta E_g$ | (1b) |
| Asymptomatic $A_g$ | $\dfrac{dA_g}{dt} = \eta\left(1-\delta\right)E_g - \mu A_g$ | (1c) |
| Infectious $I_g$ | $\dfrac{dI_g}{dt} = \eta\delta E_g - \mu I_g$ | (1d) |
| Hospitalised in acute bed $H_g$ | $\dfrac{dH_g}{dt} = \mu\gamma_g I_g - \rho H_g$ | (1e) |
| Hospitalised in IC $C_g$ | $\dfrac{dC_g}{dt} = \rho\epsilon H_g - \psi C_g$ | (1f) |
| Recovered $R_g$ | $\dfrac{dR_g}{dt} = \mu A_g + \mu\left(1-\gamma_g\right)I_g +$ $\left(1-\epsilon\right)\left(1-\kappa\right)\rho H_g +$ $\left(1-\omega_g\right)\psi C_g$ | (1g) |
| Death $D_g$ | $\dfrac{dD_g}{dt} = \left(1-\epsilon\right)\kappa\rho H_g + \omega_g\psi C_g$ | (1h) |

### Contact patterns under national interventions

We assume the population is stratified into predefined age groups with age-specific mixing pattern represented

by a contact matrix $M$ with an element of $m_{ij}$ representing the contacts between someone of age group $i \in G$ with someone of age group $j \in G$. The baseline contact matrix (with no interventions in place) is taken from the POLYMOD survey conducted in the UK.[22] The contact pattern may also be influenced by a range of interventions (social distancing was encouraged on 15 March 2020, schools were closed and lockdown occurred on 23 March 2020). We implement these interventions by assuming that the percentage of 0–18 year-olds attending school after 23 March 2020 was 5% (reducing all contacts between school-age individuals by 95%) and that social distancing reduced all contacts by 0%–50%. We take the element-wise minimum for each age group's contact with another age group from all active interventions (distancing, schools/lockdown). A study on post-lockdown contact patterns (CoMix[11]) is used to inform contacts after lockdown (first survey 24 March 2020, with an average of 73% reduction in daily contacts observed per person compared with POLYMOD).

Moving between contact matrices of multiple interventions was implemented by assuming a phased, linear decrease. After lockdown, we vary a parameter (*endphase*) to capture the time taken to fully adjust (across the population, on average) to the new measures (allowed to vary from 1 to 31 days). This assumption represents the time taken for individuals to fully adapt to new measures (and household transmission), and is in line with data on the delay in the control of COVID-19 (reductions in hospital admissions and deaths after lockdown).[23] The parameter *endphase* can be interpreted as accounting for the time taken to adjust to all interventions (and not just lockdown).

### The force of infection

The age-specific force of infection $\lambda_g$ depends on the proportion of the population who are infectious (asymptomatic $A_g$ and symptomatic $I_g$ only) and probability of transmission :

$$\lambda_g = \beta \sum_{i \in G} m_{ig} \left( \frac{A_i}{N_g} + \frac{I_i}{N_g} \right) \qquad (2)$$

### The basic reproduction number $R_0$

The basic reproduction number $R_0$ of COVID-19 is estimated to be $2.79 \pm 1.16$.[24] We include this estimate within our model by calculating the maximum eigenvalue of the contact matrix , and allowing the transmission parameter to vary such that $R_0$ is equal to the maximum eigenvalue of $M$ multiplied by the infectious period $\mu$ and the transmission parameter $\beta$. This gives the value for the initial basic reproduction number $R_0$, which changes as the contact patterns change as lockdown and other interventions are implemented.

### Parameter estimates and data sources

Model parameters are detailed in table 1. We used available published literature to inform parameter estimates. We used the following publicly available metrics for

model fitting: regional cumulative cases in SW (tested and confirmed cases in hospital), and deaths (daily/cumulative counts) from the Public Health England COVID-19 dashboard,[14] and ONS weekly provisional data on COVID-19-related deaths.[25] The case data are finalised prior to the previous 5 days, so we include all data until 14 May 2020, based on data reported until 18 May 2020. The mortality data from ONS do not explicitly state the number of COVID-19-related deaths occurring in hospital, but they do report this value nationally (83.9% of COVID-19 deaths in hospital, as of 17 April 2020). We assume that this percentage applies to the SW data and rescale the mortality to 83.9% to represent an estimate of total deaths in hospital.

### Model calibration

Using the available data (table 1), we define ranges for all parameters in our model and sample all parameters simultaneously between these minimum and maximum values assuming uniform distributions using Latin hypercube sampling (statistical method for generating random parameters from multidimensional distribution) for a total of 100 000 simulations. We used maximum likelihood estimation on total cumulative cases and cumulative deaths with a Poisson negative log likelihood calculated and summed over all observed and predicted points. For $i$ observed cases $X_i$ (from data) and $i$ predicted cases $Y_i$ (from simulations of the model), we select the best 100 parameter sets which maximise the log likelihood $\sum X_i \log (Y_i) - Y_i$ from the total sample of 100 000 simulations. The best 100 samples were taken as part of a bias–variance trade-off (online supplemental information, sensitivity analysis), and the qualitative inferences would not change with other choices of sample size. For each data point (taken from cases and deaths), we calculate this log likelihood and weight each according to the square root of the mean of the respective case or death data. This ensures that we are considering case and death data equally within our likelihood calculations.

### Model outputs

For each of the 100 best parameter sets we run the model until 11 May 2020 and output the cumulative cases and deaths in the SW. We output the predicted proportion of the population who are infectious and who have ever been infected over time. Finally, we estimate the daily and cumulative patterns of admission to and discharge from hospital (IC and acute) and cumulative mortality from COVID-19. We perform sensitivity analysis on the performance of the model when calibrated to subsets of the full data.

### RESULTS AND OUTPUTS

From 100 000 simulated parameter sets, we selected the best 100 baseline model fits on the basis of agreement to the calibration data on daily confirmed COVID-19 cases and weekly mortality due to COVID-19 in SW. The distribution of the best fitting values is shown in online

**Table 1** Parameter estimates used in the model and their sources. The distributions of unknown parameters are shown in online supplemental figure S1A for the best 100 fits

| Symbol | Description | Uniform prior (min and max) or point estimate |
|---|---|---|
| $1/\eta$ | Duration of the non-infectious exposure period | 5.1 days[41] |
| $\delta$ | Percentage of infections which become symptomatic | 82.1%[42]; vary between 73.15% and 91.05% |
| $1/\mu$ | Duration of symptoms while not hospitalised (independent of outcome) | Vary between 2 and 14 days |
| $1/\rho$ | Duration of stay in acute bed (independent of outcome) | Vary between 2 and 14 days |
| $\gamma_g$ | Percentage of symptomatic cases which will require hospitalisation | 0–4=0.00%, 5–17=0.0408%, 18–29=1.04%, 30–39=2.04%–7.00%, 40–49=2.53%–8.68%, 50–59=4.86%–16.7%, 60–69=7.01%–24.0%, 70+=9.87%–37.6%[16] |
| $1/\psi$ | Duration of stay in IC bed (independent of outcome) | 3–11 days[43] |
| $\epsilon$ | Percentage of those requiring hospitalisation who will require IC | Vary between 0% and 30% |
| $\omega_g$ | Percentage of those requiring IC who will die | 0–4=0.00%, 5–17=0.00%, 18–29=18.1%, 30–39=18.1%, 40–49=24.7%, 50–59=39.3%, 60–69=53.9%, 70+=65.3%[43] |
| $\kappa$ | Percentage of those requiring acute beds (but not IC) who will die | Vary between 5% and 35% |
| school | Percentage of 0–18 year-olds attending school after 23 March 2020 | Assume 5% |
| distancing | Percentage reduction in contact rates due to social distancing after 15 March 2020 | Vary between 0% and 50% |
| lockdown | Percentage reduction in contact rates due to lockdown after 23 March 2020 | Retail/recreation: Bristol 86%, Bath 90%, Plymouth 85%, Gloucs 84%, Somerset 82%, Devon 85%, Dorset 84%[44] Transit stations: Bristol 78%, Bath 71%, Plymouth 65%, Gloucs 69%, Somerset 67%, Devon 66%, Dorset 63%[44] Vary between 63% and 90% |
| $R_0$ | Initial reproductive number of COVID-19 | 1.63–3.95[24] |
| endphase | Time taken to fully adjust (across the population, on average) to new interventions | Vary between 1 and 31 days |

IC, intensive care.

supplemental figure S1A. All results are shown with median and 95% credible intervals (CrI).

On 11 May 2020, the reported cumulative number of individuals with (laboratory confirmed) COVID-19 was 7116 in SW,[14] and the most recent report on total cumulative deaths showed that 2306 had died from COVID-19 (as of 8 May 2020).[25]

**Estimating the total proportion of individuals with COVID-19 in SW**

Figure 2 shows the projected numbers of exposed, recovered and infectious (asymptomatic and symptomatic infections) until lockdown measures were lessened on 11 May 2020. On this date, the model predicts that a total of 5793 (95% CrI 2003 to 12 051) were infectious (0.10% of the total SW population, 95% CrI 0.04% to 0.22%). The model also predicts that a total of 189 048 (95% CrI 141 580 to 277 955) have had the virus but recovered (either asymptomatically or symptomatically), which is 3.4% (95% CrI 2.5% to 5.0%) of the SW population (not infectious and not susceptible to reinfection).

**Estimating the total hospitalised patients with COVID-19 in acute and IC beds**

The total number of patients in acute (non-IC) hospital beds across SW was projected to be 701 (95% CrI 169 to 1543) and the total number of patients in IC hospital beds was projected to be 110 (95% CrI 8 to 464) on 11 May 2020 (figure 3). Note that these ranges are quite large due to the uncertainty in the data and as more data become available these predictions will change.

**Estimating the reproduction number under interventions**

Figure 4 shows the model prediction for the reproduction ('R') number over time until 11 May 2020, when lockdown measures were relaxed. All interventions (social distancing, school closures/lockdown) had a significant impact on the reproductive number for COVID-19 in the SW. We predict that prior to any interventions R was 2.6 (95% CrI 2.0 to 3.2), and the introduction of social distancing reduced this number to 2.3 (95% CrI 1.8 to 2.9). At the minimum, R was 0.6 (95% CrI 0.5 to 0.7)

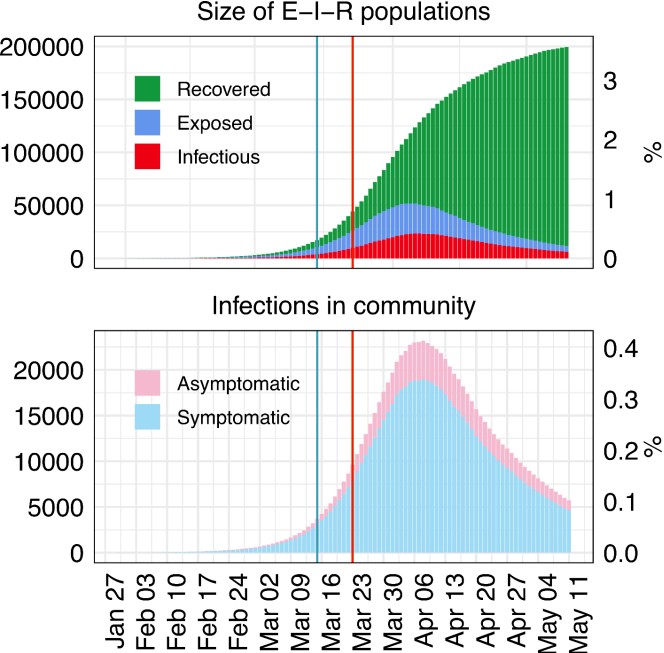

**Figure 2** The predicted median size of the exposed (E), infectious (I) and recovered (R) classes, along with the size of asymptomatic and symptomatic individuals on each day in South West England until 11 May 2020. Blue and red vertical lines represent the date the government introduced social distancing and school closures/lockdown, respectively.

after all prior interventions were enacted and adhered to (social distancing, school closures and lockdown).

Additional results for the fitting performance of the model (online supplemental figure S2A,B and table S2), the performance based on prior data (online

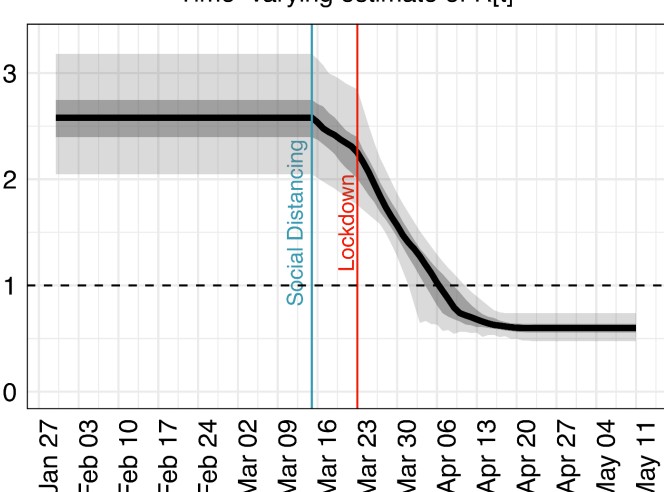

**Figure 4** The effect of interventions on estimates of R (y-axis) over time until 11 May 2020.

supplemental figure S3A–D) and sensitivity analysis can be found in the online supplemental information.

## DISCUSSION

We have developed a deterministic ordinary differential equation model of the epidemic trajectory of COVID-19 focusing on acute and IC hospital bed capacity planning to support local NHS authorities, calibrating to SW-specific data. The model is age structured and includes time-specific implementation of current interventions (following advice and enforcement of social distancing,

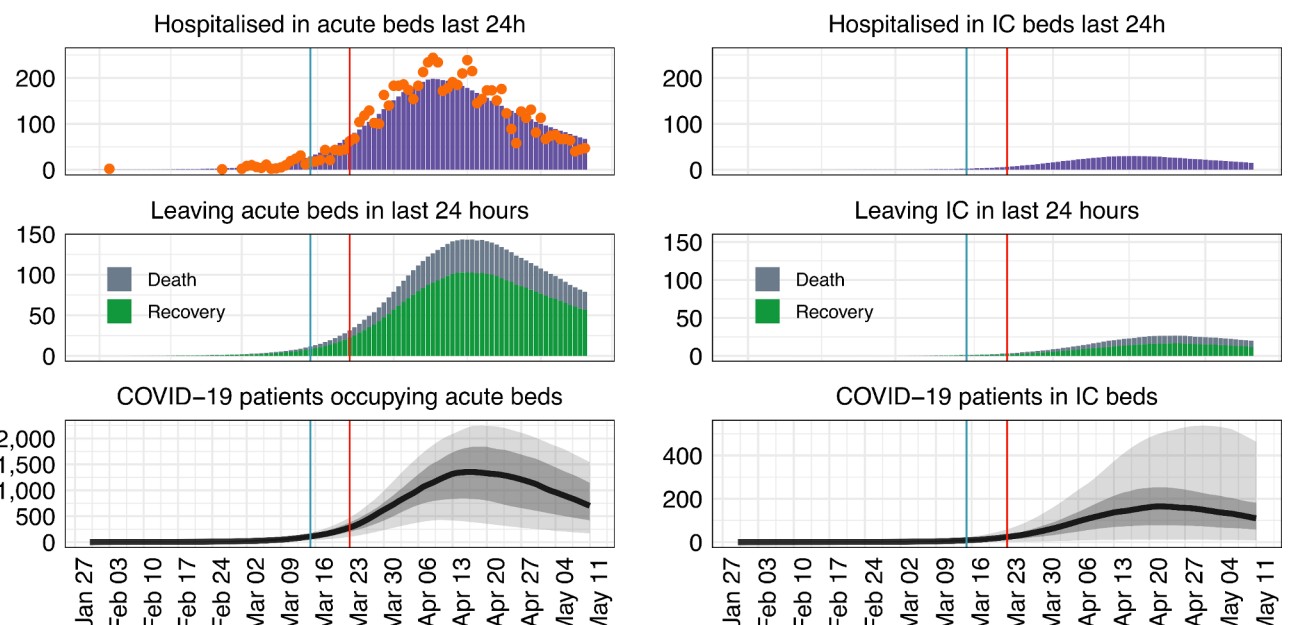

**Figure 3** The predicted number of hospitalised patients in acute and intensive care beds in the South West England (SW) until 11 May 2020. The number of daily incoming patients diagnosed with COVID-19 is shown in orange (from SW daily case data[14]), 95% credible intervals are shown in light grey, 50% in dark grey and the median value of the fits is highlighted in black. The shaded region indicates the prediction of the model from the data. Blue and red vertical lines represent the date the government introduced social distancing and school closures/lockdown, respectively. IC, intensive care.

school closures and lockdown) to predict the potential range of COVID-19 epidemic trajectories.

Using the publicly available data on cases and deaths, combined with the early estimates of parameters from early epidemics in other settings, we predict that on 11 May 2020 a total of 5793 (95% CrI 2003 to 12 051) were infectious, which equates to 0.10% (95% CrI 0.04% to 0.22%) of the total SW population. In addition, we find that the model predicts a total of 189 048 (95% CrI 141 580 to 277 955) have had the virus but recovered, which is 3.4% (95% CrI 2.5% to 5.0%) of the SW population.

We also estimate that the total number of patients in acute hospital beds in SW on 11 May 2020 was 701 (95% CrI 169 to 1543) and in IC was 110 (95% CrI 8 to 464), while the R number has decreased from 2.6 (95% CrI 2.0 to 3.2) to 0.6 (95% CrI 0.5 to 0.7) after all interventions were enacted and fully adhered to.

The fits generally agree well with both the daily case data and the cumulative count of deaths in the SW, although the model overestimates the case data at early stages and underestimates later on (which can be seen in online supplemental figure S2A, and a scatter plot of expected vs observed outputs in online supplemental figure S2B). This could be because we are using formal fitting methods or from the under-reporting of cases in the early epidemic. When assessing model performance by projecting the numbers of cases and deaths forward from four dates in April, the model performs reasonably well, with more reliable predictions occurring when more data are used to fit the model (online supplemental figure S3A–D). Even when using around half of the available data to generate forecasts (online supplemental figure S3D), the model performs reasonably well and captures the observed data later in May, but overestimates case numbers and underestimates deaths similar to those in the main analysis and in online supplemental figure S2A. This suggests that our model could perform reasonably well at predicting COVID-19 outcomes but may still slightly overestimate case numbers and underestimate deaths.

The primary strength of this study is that we have developed generalisable and efficient modelling code incorporating disease transmission, interventions and hospital bed demand which can be adapted for use in other regional or national scenarios, with the model available on GitHub for open review and use (*github.com/rdbooton/bricovmod*). We have worked closely with the NHS and at CCG level to ensure the model captures key clinical features of disease management in SW hospitals and provides output data in a format relevant to support local planning. We combined local clinical expertise with detailed literature searches to ensure reasonable parameter ranges and assumptions in the presence of high levels of parameter uncertainty.

The main challenge of this work is in balancing the urgent need locally for prediction tools which are up to date (ie, not relying on the national trends to inform capacity planning) versus more exhaustive and robust methods for model comparison. The latter of which uses existing models and more time-consuming (but more robust) data-fitting methods.[26][27] However, we believe that release of this paper and sharing of model code will facilitate multidisciplinary collaboration and rapid review and support future model comparison and uncertainty analyses.[27]

As with all models of new infections there are significant parameter uncertainties. Rapidly emerging literature is exploring a wide range of biological and epidemiological factors concerning COVID-19, but due to the worldwide nature of these studies, often parameter bands are wide and may be context specific. For example, early estimates of the basic reproduction number ranged from 1.6 to 3.8 in different locations,[28][29] with an early estimate of 2.4 used in UK model projections.[8] In addition, the information which informs our parameter selection is rapidly evolving as new data are made available, sometimes on a daily basis. From our initial analysis, we identified the following parameters as critical in determining the epidemic trajectory within our model—the percentage of infections which become symptomatic, the recovery time for cases which do not require hospital, the period between acute and IC occupancy, the length of stay in IC, the probability of transmission per contact and the gradual implementation of lockdown rather than immediate effect. Other parameters (such as the percentage reduction in school-age contacts from school closures) did not seem to influence the dynamic trajectory as strongly—and thus we assume point estimates for these parameters. However, for example, assuming that 95% of school-age contacts are reduced as a direct result of school closures is perhaps an overestimate, and future modelling work should address these uncertainties and their impacts on the epidemic trajectory of COVID-19 (but in this case, this value was somewhat arbitrary, and the assumption was used in the absence of school-age contact survey data). In addition, we did not explicitly model the societal effect prior to governmental advice (social distancing, school closures, lockdown), instead assuming a fixed date, before which we assume there were no interventions. This assumption may not be realistic and could have influenced the model output, but it is difficult to quantify the percentage compliance with interventions prior to the official release of governmental advice. More research is urgently needed to refine these parameter ranges and to validate these biological parameters experimentally. These estimates will improve the model as more empirical data become available. We look forward to reducing the uncertainty in these parameters so that we can make better predictions and fit the data more accurately.

We have also assumed that there is no nosocomial transmission of infection between hospitalised cases and healthcare workers, as we do not have good data for within-hospital transmission. However, front-line healthcare staff were likely to have been infected early on in the epidemic,[30] which could have implications for our

predicted epidemic trajectory. Our model also assumes a closed system, which may not strictly be true due to continuing essential travel. But given that up until 11 May, travel restrictions are very severe due to lockdown measures,[5] any remaining inter-regional travel is likely to have minimal effects on our model outputs. In addition, we assume that the transmission dynamics of asymptomatic individuals is equal to those of symptomatic individuals due to the viral load of asymptomatic and symptomatic carriers being comparable.[31] However, this assumption should be further explored in future modelling studies due to the potential for asymptomatic carriers to engage in higher risk behaviour and potentially impact the transmission dynamics of COVID-19.

Similar to most other COVID-19 models, we use a variant on a susceptible-exposed-infectious-recovered structure.[8–10 16 26 32 33] We do not spatially structure the population as in other UK modelling,[9 10] but we do include age-specific mixing based on POLYMOD data[22] and the postlockdown CoMix study.[11] We also explicitly measure the total asymptomatic infection, and the total in each of the clinically relevant hospital classes (acute or IC), which is a strength of our approach. Future models could also take into account local bed capacity within hospitals (including Nightingale centres) and accommodate the effect of demand outstripping supply leading to excess deaths, inclusive of non-hospital-based death such as is occurring within care homes. Future models should also address the way in which we have compartmentalised the flow of hospitalisation and death. From the symptomatic compartment, patients either recover or are admitted to hospital, from where they either die, recover or progress to IC. Under our assumption, the symptomatic recovery rate is equal to the hospitalisation rate, and the time taken for acute patients to move to IC is equal to the time to discharge for acute patients. These assumptions are a limitation of our model because in reality, those patients who progress to IC may have spent very little time in an acute bed (either due to rapid deterioration or presenting with severe symptoms). Future studies should assess the effects of these assumptions and consider other such progressions and outcomes for a patient with COVID-19 through the hospital. As with all modelling, we have not taken into account all possible sources of modelling mis-specification. Some of these mis-specifications will tend to increase the predicted epidemic period, and others will decrease it. One factor that could significantly change our predicted epidemic period is the underlying structure within the population leading to heterogeneity in the average number of contacts under lockdown, for example, key workers have high levels of contact but others are able to minimise contacts effectively, this might lead to an underestimate of ongoing transmission, but potentially an overestimate of the effect of releasing lockdown. We also know that there are important socioeconomic considerations in determining people's ability to stay at home and particularly to work from home.[34]

Early UK modelling predicted the infection peak to be reached roughly 3 weeks from the initiation of severe lockdown measures, as taken by the UK government in mid-March.[8] A more recent study factoring spatial distribution of the population indicated the peak to follow in early April due to $R_0$ reducing to below 1 in many settings in weeks following lockdown.[9] Other modelling indicated that deaths in the UK would peak in mid-late April; furthermore, that the UK would not have enough acute and IC beds to meet demand.[35] While modelling from the European Centre for Disease Prevention and Control estimated peak cases to occur in most European countries in mid-April,[20] these estimations were largely at a national level. Due to the expected lag of other regions behind London, these estimated peaks are likely to be shifted further into the future for the separate regions of the UK, and as shown by our model occurred in early to mid-April. This is also likely to be true for future peaks which may result from relaxing lockdowns.

Outside of the UK, a similar modelling from France[32] (which went into lockdown at a similar time the UK on 17 March) predicted the peak in daily IC admissions at the end of March. Interestingly, however, when dissected by region, the peak in IC bed demand varied by roughly 2 weeks. Swiss modelling similarly predicted a peak in hospitalisation and numbers of patients needing IC beds in early April, after lockdown implementation commenced on 17 March.[33] US modelling[36] disaggregated by State also highlights the peak of excess bed demand varies geographically, with this peak ranging from the second week of April through to May, dependent on the State under consideration. The modelling based in France also cautioned that due to only 5.7% of the population having been infected by 11 May when the restrictions would be eased, the population would be vulnerable to a second epidemic peak thereafter.[32]

The ONS in England estimated that an average of 0.25% of the population had COVID-19 between 4 and 17 May 2020 (95% CI 0.16% to 0.38%),[37] which is greater than the 0.10% (95% CrI 0.04% to 0.22%) we found with our model (on 11 May 2020), but with some overlap. In addition, the ONS estimated that 6.78% (95% CrI 5.21% to 8.64%) tested positive for antibodies to COVID-19 up to 24 May 2020 in England,[38] and Public Health England estimated that approximately 4% (2%–6%) tested positive for antibodies to COVID-19 between 20 and 26 April 2020 in the SW.[39] Compared with our model, 3.4% (95% CrI 2.5% to 5.0%) had recovered on 11 May 2020 (2 weeks later), demonstrating that our model estimates may be within sensible bounds, and further highlighting the need for more regional estimates of crucial epidemiological parameters and seroprevalence. We have assumed that individuals are not susceptible to reinfection within the model time frame; however, in future work it will be important to explore this assumption. It is not known what the long-term pattern of immunity to COVID-19 will be,[40] and this will be key to understanding the future

epidemiology in the absence of a vaccine or effective treatment options.

With this in mind, our findings demonstrate that there are still significant data gaps—and in the absence of such data, mathematical models can provide a valuable asset for local and regional healthcare services. This regional model will be used further in the SW as the pandemic evolves and could be used within other healthcare systems in other geographies to support localised predictions. Controlling intervention measures at a more local level could be made possible through monitoring and assessment at the regional level through a combination of clinical expertise and local policy, guided by localised predictive forecasting as presented in this study.

**Author affiliations**
[1]School of Veterinary Sciences, University of Bristol, Bristol, UK
[2]Population Health Science Institute, University of Bristol Medical School, Bristol, UK
[3]NIHR Health Protection Research Unit in Behavioural Science and Evaluation, Bristol, UK
[4]Academic Respiratory Unit, Southmead Hospital, Bristol, UK
[5]Immunology, Pathology Sciences, North Bristol NHS Trust, Bristol, UK
[6]Consultant in Microbiology, University Hospitals Bristol, Bristol, UK
[7]Consultant in Microbiology and Infectious Diseases, University Hospitals Bristol, Bristol, UK
[8]Infection Science, Southmead Hospital, North Bristol NHS Trust, Bristol, UK
[9]School of Mathematics, University of Bristol, Bristol, UK
[10]Department of Engineering Mathematics, University of Bristol, Bristol, UK
[11]Alan Turing Institute, London, UK
[12]Health Data Research UK South-West of England Partnership, Bristol, UK
[13]Modelling and Analytics Team, NHS Bristol, North Somerset and South Gloucestershire CCG, Bristol, UK

**Contributors** Conception and design of the study: LD, EBP. Acquisition of data: RDB, EBP, RW, KMET. Mathematical modelling: RDB, LM, LD, EBP, RMW, KMET. Coding and simulations: RDB. Analysis and interpretation of results: RDB, LM, LV, KJL, CH, PDB, IH, RL, FH, DL, LD, AP, EBP, RMW, KMET. Writing and drafting of the manuscript: RDB, LM, LV, KJL, CH, PDB, IH, RL, FH, DL, LD, AP, EBP, RMW, KMET. Approval of the submitted manuscript: RDB, LM, LV, KJL, CH, PDB, IH, RL, FH, DL, LD, AP, EBP, RMW, KMET.

**Funding** This work was supported by Global Public Health strand of the Elizabeth Blackwell Institute for Health Research, funded under the University of Bristol's QR GCRF strategy (award number ISSF3: 204813/Z/16/Z). This work was also funded with support from Bristol UNCOVER (Bristol COVID Emergency Research, award number ISSF3: 204813/Z/16/Z) and Medical Research Council UK (award number MR/S004769/1). LM, KJL, EBP and KMET acknowledge the support from the NIHR Health Protection Research Unit in Behavioural Science and Evaluation at the University of Bristol (award number NIHR200877). This work was also supported by Health Data Research UK, which is funded by the UK Medical Research Council, Engineering and Physical Sciences Research Council, Economic and Social Research Council, National Institute for Health Research, Chief Scientist Office of the Scottish Government Health and Social Care Directorates, Health and Social Care Research and Development Division (Welsh Government), Public Health Agency (South Western Ireland), British Heart Foundation and Wellcome (award number CFC0129).

**Competing interests** None declared.

**Patient consent for publication** Not required.

**Provenance and peer review** Not commissioned; externally peer reviewed.

**Data availability statement** Data are available in a public, open access repository. All data relevant to the study are included in the article or uploaded as supplementary information. All model code are open source and available for download on GitHub: https://github.com/rdbooton/bricovmod. All data are freely available via the GOV.UK COVID-19 dashboard and ONS.

**ORCID iDs**
Ross D Booton http://orcid.org/0000-0002-3013-4179
Katharine J Looker http://orcid.org/0000-0002-3375-0807
Daniel Lawson http://orcid.org/0000-0002-5311-6213
Richard Wood http://orcid.org/0000-0002-3476-395X
Ellen Brooks-Pollock http://orcid.org/0000-0002-5984-4932
Katherine M E Turner http://orcid.org/0000-0002-8152-6017

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
