## [Reviewer comments · BMJ Open]

ARTICLE DETAILS

TITLE (PROVISIONAL)	Estimating the COVID-19 epidemic trajectory and hospital capacity requirements in South West England: a mathematical modelling framework
AUTHORS	Booton, Ross; MacGregor, Louis; Vass, Lucy; Looker, Katharine; Hyams, Catherine; Bright, Philip; Harding, Irasha; Lazarus, Rajeka; Hamilton, Fergus; Lawson, Daniel; Danon, Leon; Pratt, Adrian; Wood, Richard; Brooks-Pollock, Ellen; Turner, Katy

VERSION 1 – REVIEW

REVIEWER	David Scheinker Stanford University, United States My research group has developed models of COVID-19 hospital demand for the United States. Our models are of a sufficiently different kind and purpose that my work on them does not present a significant competing interest.
REVIEW RETURNED	28-Jul-2020

GENERAL COMMENTS	Review of Estimating the COVID-19 epidemic trajectory and hospital capacity requirements in South West England: a mathematical modelling framework In this work the authors model the spread of COVID-19 in South West England with a susceptible-exposed-infectious-recovered (SEIR) compartmental model parameterized with estimates from the medical literature and national tracking programs. The work is thoughtfully designed, clearly written, and well executed. Several strengths of the work suggest that it may contribute to efforts to understand, prepare a response, and design strategies to control the spread of the disease. The model explicitly captures two important features of COVID-19 that differentiate it from other influenza epidemics: a significant percentage of asymptomatic infectious individuals and significant differences in age-specific risk of hospitalization or death. The model accommodates different mixing rates between age groups and age-specific results of non-pharmaceutical interventions such as school closures. The model is designed to facilitate decision making at the local level, particularly about estimates of the numbers of hospital acute care and ICU beds necessary to accommodate COVID-19 patients. This methodological choice makes the results more useful to hospital and regional decision makers than the results of national models. The data and code are made available for other researchers and are relatively well documented. The work includes an extensive sensitivity analysis of model parameters within bounds based on
---

	empirical data. This is an important methodological step, but the most meaningful improvements will likely come from incorporating better estimates into the model as more empirical data become available. I have three requests for revisions somewhere between minor and major:  1. Numerous SEIR models of COVID-19 have been published. A summary of the most relevant models as well as more detailed description of how this model complements those models would be very helpful to contextualize the contribution of the present work. 2. The model has two potential uses. A better retrospective analysis of disease transmission and as a tool to forecast disease spread. As part of the variance-bias tradeoff during model calibration, the authors use the 100 top-performing sets of parameters in order to avoid overfitting the data - a reasonable but necessarily arbitrary choice examined with sensitivity analyses. A more interpretable approach to evaluate the model would be to choose the parameters based on only the first 80% of the available data, perhaps through mid-April, and then evaluate model performance by projecting the numbers of infections forward through April 11 and comparing these projects to the held-out data that had not been used to calibrate the model. This would produce higher errors, but would give readers an interpretable way to estimate how well the forecasts generated by the model could be expected to perform. 3. While the inclusion of asymptomatic patients is an important strength of the model, the assumption that their transmission dynamics are the same as those of symptomatic patients is not discussed. There are two factors that should be discussed further or potentially incorporated into the force of transmission as parameters specific to the asymptomatic group: asymptomatic individuals may be nearly as infectious as symptomatic individuals but may engage in higher risk behavior. Numerous reports present data and analyze transmission by asymptomatic individuals, e.g., Huff, H.V. and Singh, A., 2020. Asymptomatic transmission during the COVID-19 pandemic and implications for public health strategies. Clinical Infectious Diseases and Payne, D.C., Smith-Jeffcoat, S.E., Nowak, G., Chukwuma, U., Geibe, J.R., Hawkins, R.J., Johnson, J.A., Thornburg, N.J., Schiffer, J., Weiner, Z. and Bankamp, B., 2020. SARS-CoV-2 Infections and Serologic Responses from a Sample of US Navy Service Members—USS Theodore Roosevelt, April 2020. Morbidity and Mortality Weekly Report, 69(23), p.714. Three minor comments:  1. Comparison of the model with observed data should be flushed out with further details. The authors provide two visual representations - Figures S2a & S2b. It should be complemented with more details in a table in the appendix. 2. The results of the sensitivity analysis should be expanded to include peak Total Acute Beds and Peak Total ICU Beds in addition to total beds. 3. Parts of the documentation of the code could be revised to be more precise (e.g., line 162 - "#schools open - assume 50% went back???" You will need to find values for this later")
--	--

REVIEWER	Tjibbe Donker University Medical Center Freiburg, Germany
REVIEW RETURNED	07-Aug-2020

GENERAL COMMENTS	Booton et al show how deterministic modelling together with local
---

data can be used to predict local or regional healthcare demand in a pandemic situation. The focus on a local or regional ability to estimate key healthcare demands is a crucially important one, as the COVID-19 pandemic displays a distinct regional dynamic.

Hospitalisations and other healthcare demand can therefore differ widely between regions, and national forecasts may be of little use to specific regions. The manuscript is well written, the model well-structured and clear, and I enjoyed reading

There is one major difficulty I run into with the manuscript, and it revolves primarily around the presentation of the goal of the model. The last sentence of the discussion says: "...guided by localised predictive forecasting as presented in this study".

But if I understand correctly, the model doesn't produce a forward prediction (forecast), but a prediction of current hospital capacity requirements, given the epidemic trajectory up to today. Specifically, the model parameters are fitted, using Latin Hypercube Sampling, to data of case counts and reported deaths until May 14 (line 205), and the simulations run until May 11 (Line 230). Any prediction is thus made over the period for which case data was available. The model output on bed demand (Acute beds and ICU) are then intermediate steps between the fitted case and death data, and indeed form a prediction. The model parameters do enable forecasting (in particular on the short term), but the authors did not pursue this avenue.

I'm not saying the authors should try (short-term) forecasting, but they should present the strengths of this particular approach better. In my view, the prediction of local bed demand is a very useful tool, even if the predictions are only produced until today. If the author were to present this as a forward prediction, they should at least show if the model can predict further than the data it is fitted on. This can be done by fitting the model to earlier data than May 11, or evaluating the past model runs to data that was gathered since May 11.

The deterministic model has some drawbacks that should be addressed. For instance the way hospitalisation and death is modelled. From the symptomatic compartment (I), patients either recover or are admitted to hospital (H), from where they die, recover, or move on to the ICU. The proportions of patients going to each next compartment in this example are determined by gamma and epsilon. Because of the assumption of constant rates, this would mean that on average patient being hospitalised was just as long symptomatic (before hospitalisation) as a patient that recovered. The same goes for those moving to ICU: they have spent just as much time in hospital as those being discharged.

Again, although I understand the simplifying assumption, it may affect the final results. In reality, a lot of patients who end up on ICU with COVID spent very little time on the general ward (due to rapid deterioration).

Minor points:

- Should endphase be given in table 1?
- figure s1b (uniform priors) is uninformative, and the priors are already given in table one. I'd suggest leaving the figure out.
- the time of implementation of social distancing and lockdown seem to be fixed, causing the stable R estimate before social distancing as visible in figure 4. However, there might have been some societal effect prior to the governments. Did you consider allowing some

VERSION 1 – AUTHOR RESPONSE

Reviewer(s)' Comments to Author:

Reviewer: David Scheinker, Stanford University, United States. *Competing interests: My research group has developed models of COVID-19 hospital demand for the United States. Our models are of a sufficiently different kind and purpose that my work on them does not present a significant competing interest.*

1. In this work the authors model the spread of COVID-19 in South West England with a susceptible-exposed-infectious-recovered (SEIR) compartmental model parameterized with estimates from the medical literature and national tracking programs. The work is thoughtfully designed, clearly written, and well executed. Several strengths of the work suggest that it may contribute to efforts to understand, prepare a response, and design strategies to control the spread of the disease. The model explicitly captures two important features of COVID-19 that differentiate it from other influenza epidemics: a significant percentage of asymptomatic infectious individuals and significant differences in age-specific risk of hospitalization or death.

Reply: We thank the reviewer for their careful evaluation of our work. We appreciate their positive comments, and suggestions for improvements.

2. The model accommodates different mixing rates between age groups and age-specific results of non-pharmaceutical interventions such as school closures. The model is designed to facilitate decision making at the local level, particularly about estimates of the numbers of hospital acute care and ICU beds necessary to accommodate COVID-19 patients. This methodological choice makes the results more useful to hospital and regional decision makers than the results of national models.

Reply: The reviewer is correct that the results may be more useful to regional decision makers and hospitals rather than at the national level. We are particularly glad that this message was communicated clearly to the reviewer in our manuscript.

3. The data and code are made available for other researchers and are relatively well documented. The work includes an extensive sensitivity analysis of model parameters within bounds based on empirical data. This is an important methodological step, but the most meaningful improvements will likely come from incorporating better estimates into the model as more empirical data become available.

Reply: We thank the reviewer for their comments on the open data and code we have supplied alongside our manuscript. We completely agree with the reviewer on their second point and amended the manuscript as follows:

Page 19, Lines 383-387 “More research is urgently needed to refine these parameter ranges and to validate these biological parameters experimentally. These estimates will improve the model as more empirical data becomes available. We look forward to reducing the uncertainty in these parameters so that we can make better predictions and fit the data more accurately.”

4. I have three requests for revisions somewhere between minor and major: Numerous SEIR models of COVID-19 have been published. A summary of the most relevant models as well as more detailed description of how this model complements those models would be very helpful to contextualize the contribution of the present work.

Reply: We thank the reviewer for this suggestion which will greatly improve the overall readability of the introduction and will place our work within the wider literature. Therefore, we have amended the second paragraph of the introduction to include the suggested summary of relevant models:

Pages 4 and 5 Lines 83-100 “Mathematical modelling has been used to predict the course of the COVID-19 pandemic and to evaluate the effectiveness of proposed and enacted interventions [6–11]. Prem et al. [6] showed that the premature lifting of control strategies at the national level (within China) could lead to an earlier secondary peak; Flaxman et al. [7] used a semi-mechanistic model to predict the total COVID-19 infections in 11 countries; Ferguson et al. [8] used an individual-based simulation model of COVID-19 transmission to explore the effects of non-pharmaceutical interventions within the USA and Great Britain; Challen et al. [9] estimated the R-number among regions of the UK; Danon et al. [10] used a spatial model to predict the potential course of COVID-19 in England and Wales in the absence of control measures; while Jarvis et al. [11] analysed the behavioural monitoring data to quantify the impact of control measures on COVID-19 transmission. These models have been predominantly aimed at the national level and have largely been based on epidemiological and biological data sourced from the initial epidemic in Wuhan, China [12] and the first large outbreak in Lombardy, Italy [13]. These models have also mainly focused on predicting the scale of COVID-19 transmission under various intervention measures, rather than producing estimates for potential numbers of COVID-19 related admissions to acute or intensive care.”

5. The model has two potential uses. A better retrospective analysis of disease transmission and as a tool to forecast disease spread. As part of the variance-bias tradeoff during model calibration, the authors use the 100 top-performing sets of parameters in order to avoid overfitting the data - a reasonable but necessarily arbitrary choice examined with sensitivity analyses. A more interpretable approach to evaluate the model would be to choose the parameters based on only the first 80% of the available data, perhaps through mid-April, and then evaluate model performance by projecting the numbers of infections forward through April 11 and comparing these projects to the held-out data that had not been used to calibrate the model. This would produce higher errors, but would give readers an interpretable way to estimate how well the forecasts generated by the model could be expected to perform.

Reply: We thank the reviewer for this valuable suggestion to evaluate the performance of the model based on previous data. We have added an entire section to the supplement “Model performance”, where we adapt Figure S2a to show the model performance using only data until 4 dates in April a) 27th April 2020, b) 20th April 2020, c) 13th April 2020 and d) 6th April 2020). This represents 82%, 72%, 62% and 53% of the available observed data points. The model performs reasonably well, with final actuals contained within the 95% credible interval regardless of date selected, however with more reliable predictions occurring with the fits using the most data.

We perform out-of-sample model verification by reporting performance of the best 100 simulations with different observation windows, which captures predictive performance as the reviewer suggests. We note that it would be appealing to use this approach to estimate how many simulations to retain, but this is not straightforward. Due to the correlations within our data, there is only one way to reserve test data - after a specific date. Maximising test-set performance would not fully address uncertainty: the number of simulations retained needs to be calibrated not just on their predictive performance on what really happened after this date, but on the whole distribution of what may have happened. In light of this, we added the following section to the supplement, added new figures to illustrate the reviewer’s suggestions, and added a paragraph to the discussion

Supplement Pages 10-11 Lines 84-104 “Here we evaluate the model performance as a tool to forecast the potential cumulative cases and deaths related to COVID-19. We do this by using only a certain percentage of the total available data until a) 27th April 2020, b) 20th April 2020, c) 13th April 2020 and d) 6th April 2020) to fit the model, then comparing these projections to the actual data observed past each of these dates. This gives an interpretable way to estimate how well the forecast generated by the model could be expected to perform.

Figures S3a-d show the overall performance of the model, which in general overestimates the total cumulative cases, but captures the cumulative deaths more accurately. All final estimates from the data are within the 95% expected credible interval predictions regardless of the date chosen to fit the model to.

Figure S3a shows the model prediction using data only up to 27th April 2020, which is based on 82% of the available data. This the most reliable prediction, while the most variable predictions come from those which use 53% of the available data (Figure S3d), suggesting that the model can accurately

forecast the immediate future with higher reliability – while further time points are harder to predict (although the model predictions based on only 53% of the data also perform reasonably well and the final actuals are within the predicted 95% credible interval).”

Supplement Page 11-14 - Figure S3a, Figure S3b, Figure S3c and Figure S3d.

Page 17 Lines 332-340 “When assessing model performance by projecting the numbers of cases and deaths forward from four dates in April, the model performs reasonably well, with more reliable predictions occurring when more data is used to fit the model (Figures S3a, S3b, S3c and S3d). Even when using around half of the available data to generate forecasts (Figure S3d), the model performs reasonably well and captures the observed data later in May, but overestimates case numbers and underestimates deaths similar to those in the main analysis and in Figure S2a. This suggests that our model could perform reasonably well at predicting COVID-19 outcomes but may still slightly overestimate case numbers and underestimate deaths.”

6. While the inclusion of asymptomatic patients is an important strength of the model, the assumption that their transmission dynamics are the same as those of symptomatic patients is not discussed. There are two factors that should be discussed further or potentially incorporated into the force of transmission as parameters specific to the asymptomatic group: asymptomatic individuals may be nearly as infectious as symptomatic individuals but may engage in higher risk behavior. Numerous reports present data and analyze transmission by asymptomatic individuals, e.g., Huff, H.V. and Singh, A., 2020. Asymptomatic transmission during the COVID-19 pandemic and implications for public health strategies. Clinical Infectious Diseases and Payne, D.C., Smith-Jeffcoat, S.E., Nowak, G., Chukwuma, U., Geibe, J.R., Hawkins, R.J., Johnson, J.A., Thornburg, N.J., Schiffer, J., Weiner, Z. and Bankamp, B., 2020. SARS-CoV-2 Infections and Serologic Responses from a Sample of US Navy Service Members—USS Theodore Roosevelt, April 2020. Morbidity and Mortality Weekly Report, 69(23), p.714.

Reply: The comment from the reviewer is a very interesting and important point – we agree that the behaviour of asymptomatic individuals would indeed be different and agree that future studies should look at the effects of this (and possible interventions?). Looking at the higher risk behaviour of these individuals could be a really interesting extension to our work. We appreciate this comment and have added a paragraph to the discussion with suggested reference to Huff, H.V. and Singh, A., 2020.

Page 19 Lines 395-400 “In addition, we assume that the transmission dynamics of asymptomatic individuals is equal to those of symptomatic individuals, due to the viral load of asymptomatic and symptomatic carriers being comparable [35]. However, this assumption should be further explored in future modelling studies due to the potential for asymptomatic carriers to engage in higher-risk behaviour, and potentially impact the transmission dynamics of COVID-19.”

7. Three minor comments: Comparison of the model with observed data should be flushed out with further details. The authors provide two visual representations - Figures S2a & S2b. It should be complemented with more details in a table in the appendix.

Reply: We thank the reviewer for this comment and agree with their suggestion to complement the figures in the supplement with a table in the appendix. This table can be found in the supplement:

Supplement Pages 6-9: Table S2: The predicted median and 95% credible interval for cumulative cases and deaths alongside the observed cumulative cases and deaths.

8. The results of the sensitivity analysis should be expanded to include peak Total Acute Beds and Peak Total ICU Beds in addition to total beds.

Reply: We thank the reviewer for their minor suggestion on our sensitivity analysis for total recovered, infectious, total in acute beds and IC beds. Our sensitivity analysis was performed to highlight the difference in the main outcomes of our model (bed modelling in hospital, or total infectious/recovered) in response to the size of the final sample size taken from 100k samples. This enables us to show the reader our bias-variance trade off and show that qualitative inferences would not change with other choices of sample size. Therefore, the sensitivity analysis should not be expanded in this way, and do not think this would add any more detail to the purpose of this analysis described above.

9. Parts of the documentation of the code could be revised to be more precise (e.g., line 162 -

"#schools open - assume 50% went back??? You will need to find values for this later")

Reply: We thank the reviewer for spotting this (we should have originally removed this comment from the code). We have now removed the comment on line 162 from the online GitHub repository. We have uploaded a new bricovmodV3.R file to <https://github.com/rdbooton/bricovmod>

Reviewer: Tjibbe Donker, University Medical Center Freiburg, Germany. Competing interests:
None declared

1. Booton et al show how deterministic modelling together with local data can be used to predict local or regional healthcare demand in a pandemic situation. The focus on a local or regional ability to estimate key healthcare demands is a crucially important one, as the COVID-19 pandemic displays a distinct regional dynamic. Hospitalisations and other healthcare demand can therefore differ widely between regions, and national forecasts may be of little use to specific regions. The manuscript is well written, the model well-structured and clear, and I enjoyed reading

Reply: We thank the reviewer for their positive evaluation of our manuscript and for their careful comments below which has improved our manuscript.

2. There is one major difficulty I run into with the manuscript, and it revolves primarily around the presentation of the goal of the model. The last sentence of the discussion says: "...guided by localised predictive forecasting as presented in this study". But if I understand correctly, the model doesn't produce a forward prediction (forecast), but a prediction of current hospital capacity requirements, given the epidemic trajectory up to today. Specifically, the model parameters are fitted, using Latin Hypercube Sampling, to data of case counts and reported deaths until May 14 (line 205), and the simulations run until May 11 (Line 230). Any prediction is thus made over the period for which case data was available. The model output on bed demand (Acute beds and ICU) are then intermediate steps between the fitted case and death data, and indeed form a prediction. The model parameters do enable forecasting (in particular on the short term), but the authors did not pursue this avenue. I'm not saying the authors should try (short-term) forecasting, but they should present the strengths of this particular approach better. In my view, the prediction of local bed demand is a very useful tool, even if the predictions are only produced until today. If the author were to present this as a forward prediction, they should at least show if the model can predict further than the data it is fitted on. This can be done by fitting the model to earlier data than May 11, or evaluating the past model runs to data that was gathered since May 11.

Reply: We thank the reviewer for their suggestions, which align well with reviewer 1 point 5. Because of this, we present a new analysis in the supplementary analysis. We now fit the model to earlier data over four weeks in April 2020: a) 27th April 2020, b) 20th April 2020, c) 13th April 2020 and d) 6th April 2020 to explore the potential for our model to predict the trajectory of future COVID-19 outcomes. The model performs relatively well, and we have added a new paragraph to the discussion section of the main text, as well as an entire section with more detail in the supplement with accompanying figures S3a, S3b, S3c and S3d. We appreciate both reviewer's comments on model performance which will aid the discussion section of our manuscript.

Responses can be found in point 5 reviewer 1

Page 17 Lines 332-340 "When assessing model performance by projecting the numbers of cases and deaths forward from four dates in April, the model performs reasonably well, with more reliable predictions occurring when more data is used to fit the model (Figures S3a, S3b, S3c and S3d). Even when using around half of the available data to generate forecasts (Figure S3d), the model performs reasonably well and captures the observed data later in May, but overestimates case numbers and underestimates deaths similar to those in the main analysis and in Figure S2a. This suggests that our model could perform reasonably well at predicting COVID-19 outcomes but may still slightly overestimate case numbers and underestimate deaths."

3. The deterministic model has some drawbacks that should be addressed. For instance the way hospitalisation and death is modelled. From the symptomatic compartment (I), patients either recover or are admitted to hospital (H), from where they die, recover, or move on to the ICU. The proportions of patients going to each next compartment in this example are determined by gamma and epsilon. Because of the assumption of constant rates, this would mean that on average patient being hospitalised was just as long symptomatic (before hospitalisation) as a patient that recovered. The

same goes for those moving to ICU: they have spent just as much time in hospital as those being discharged. Again, although I understand the simplifying assumption, it may affect the final results. In reality, a lot of patients who end up on ICU with COVID spent very little time on the general ward (due to rapid deterioration).

Reply: This is a very good suggestion – we agree with the reviewer on this and have added a paragraph to the discussion to explain these assumptions and present these as a limitation of our approach:

Page 20 Lines 410-420 “Future models should also address the way in which we have compartmentalised the flow of hospitalisation and death. From the symptomatic compartment, patients either recover or are admitted to hospital, from where they either die, recover or progress to IC. Under our assumption, the symptomatic recovery rate is equal to the hospitalisation rate, and the time taken for acute patients to move to IC is equal to the time to discharge for acute patients. These assumptions are a limitation of our model, because in reality those patients who progress to IC may have spent very little time in an acute bed (either due to rapid deterioration or presenting with severe symptoms). Future studies should assess the effects of these assumptions and consider other such progressions and outcomes for a COVID-19 patient through the hospital.”

4. Minor points: Should endphase be given in table 1?

Reply: We have updated Table 1 to include the parameter *endphase*.

5. figure s1b (uniform priors) is uninformative, and the priors are already given in table one. I'd suggest leaving the figure out.

Reply: We agree with the reviewer. We removed Figure S1b from the supplement and any references to it in the main text.

6. the time of implementation of social distancing and lockdown seem to be fixed, causing the stable R estimate before social distancing as visible in figure 4. However, there might have been some societal effect prior to the governments. Did you consider allowing some variation in this date (or these dates).

Reply: We did not consider allowing variation in the social distancing (15th March 2020) and the lockdown date (23rd March 2020). But this is an interesting idea, we thank the reviewer for suggesting this possible extension to our model. It would be extremely difficult to quantify this and would only add to the uncertainty in the model, and in the absence of data we have added a sentence to the discussion to highlight this point.

Page 19 Lines 378-387 “In addition, we did not explicitly model the societal effect prior to governmental advice (social distancing, school closures, lockdown), instead assuming a fixed date, before which we assume there was no interventions. This assumption may not be realistic and could have influenced the model output, but it is difficult to quantify the percentage compliance with interventions prior to the official release of governmental advice.”

VERSION 2 – REVIEW

REVIEWER	David Scheinker Stanford University, United States My research group has developed models of COVID-19 hospital bed demand. Our models are of a very different kind and designed for a very different purpose. I don't think my work presents a significant competing interest.
REVIEW RETURNED	21-Sep-2020
GENERAL COMMENTS	Thank you for addressing all suggestions.
REVIEWER	Tjibbe Donker

	University Medical Center Freiburg, Freiburg im Breisgau, Germany
REVIEW RETURNED	20-Oct-2020

GENERAL COMMENTS	The authors have thoroughly addressed my comments and questions, as well as those questions raised by the other reviewer. The revised version has improved considerably, and reads very well. I have no further comments.
---